# Venetoclax Resistance in Acute Myeloid Leukemia

**DOI:** 10.3390/cancers16061091

**Published:** 2024-03-08

**Authors:** Sylvain Garciaz, Marie-Anne Hospital, Yves Collette, Norbert Vey

**Affiliations:** Aix-Marseille University, Inserm, CNRS, Institut Paoli-Calmettes, CRCM, 13009 Marseille, France; hospitalm@ipc.unicancer.fr (M.-A.H.); collettey@ipc.unicancer.fr (Y.C.); veyn@ipc.unicancer.fr (N.V.)

**Keywords:** venetoclax, acute myeloid leukemia, apoptosis

## Abstract

**Simple Summary:**

Venetoclax–azacitidine is a new standard for elderly or unfit acute myeloid leukemia patients. Nevertheless, resistance remains a matter of concern. The main genetic alterations associated with venetoclax sensitivity are IDH mutations, whereas TP53, signaling mutations, and BAX mutations are associated with venetoclax resistance. Non-genetic resistance mechanisms have also been described, including changes in apoptotic proteins, differentiation states, metabolic status, and mitochondrial machinery. Venetoclax-based triplet therapies including IDH and FLT3 inhibitors or innovative therapies are under investigation to target resistances.

**Abstract:**

Venetoclax is a BH3-mimetics agent interacting with the anti-apoptotic protein BCL2, facilitating cytochrome c release from mitochondria, subsequent caspases activation, and cell death. Venetoclax combined with azacitidine (VEN-AZA) has become a new standard treatment for AML patients unfit for intensive chemotherapy. In the phase III VIALE-A study, VEN-AZA showed a 65% overall response rate and 14.7 months overall survival in comparison with 22% and 8 months in the azacitidine monotherapy control arm. Despite these promising results, relapses and primary resistance to venetoclax are frequent and remain an unmet clinical need. Clinical and preclinical studies have been conducted to identify factors driving resistance. Among them, the most documented are molecular alterations including *IDH*, *FLT3*, *TP53*, and the newly described *BAX* mutations. Several non-genetic factors are also described such as metabolic plasticity, changes in anti-apoptotic protein expression, and dependencies, as well as monocytic differentiation status. Strategies to overcome venetoclax resistance are being developed in clinical trials, including triplet therapies with targeted agents targeting IDH, FLT3, as well as the recently developed menin inhibitors or immunotherapies such as antibody–drug conjugated or monoclonal antibodies. A better understanding of the molecular factors driving venetoclax resistance by single-cell analyses will help the discovery of new therapeutic strategies in the future.

## 1. Introduction

The combination of the BCL2 inhibitor venetoclax (VEN) and azacitidine (AZA) has changed the paradigm of treatment for elderly or unfit acute myeloid leukemia (AML) patients. Indeed, in this hard-to-treat population, the use of hypomethylating agents (HMAs) such as decitabine or AZA was associated with a 20–30% complete response rate and 10 months survival [1,2]. In comparison, response rates with VEN-AZA are approximately 65%, and overall survival rate was 14.7 months in the pivotal VIALE-A study [3]. The long-term analyses of patients treated in this study have been recently updated with a 2-year OS of 37.5%, representing a gain of 20.6% compared to AZA monotherapy [4]. Nonetheless, primary resistance to VEN-AZA represents approximately one-third of treated AML patients. Moreover, the median duration of remission for responding patients is approximately 18 months, and virtually all the patients will finally relapse, as shown in the absence of a “plateau” on the survival curves, indicating that the regimen is not a curative therapy.

VEN-AZA relapsed or refractory AML (R/R-AML) patients have very poor outcomes, with a response rate and OS of about 20% and 2.4 months, respectively [5,6]. Therefore, a better understanding of VEN resistance is crucial to developing new therapeutic strategies and improving the survival of VEN-AZA R/R AML [6]. Recent reviews have pinpointed the role of genetic and non-genetic factors in the apparition of resistance [7,8,9,10,11,12,13,14]. In this review, we will describe the main mechanisms of resistance and the therapeutic strategies to overcome them. Finally, we will describe the most promising triplet therapies using VEN-AZA as a backbone for the treatment of unfit/elderly AML patients.

## 2. Rationale of BCL2 Inhibition

### 2.1. Mechanism of Intrinsic Apoptosis and Developmlent of BH3 Mimetics

Intrinsic apoptosis relies on the balance between pro- and anti-apoptotic proteins inducing mitochondrial outer membrane permeabilization (MOMP), ultimately leading to caspase-dependent cell death. MOMP is regulated by the members of the BCL2 protein family containing a single BH3 domain named “BH3-only proteins”, which have a pro-apoptotic role. Conversely, MOMP is blocked by a series of proteins that have an anti-apoptotic action, including BCL2, BCL-XL, and MCL1. When the balance between pro- and anti-apoptotic signals turns toward cell death due to internal or external stressors, the BAX and BAK proteins are activated and form pores in the mitochondrial outer membrane, subsequently releasing cytochrome c and inducing caspase activation [12].

Efforts have been made to develop BH3-mimetic agents that induce intrinsic apoptosis by blocking anti-apoptotic proteins. The first synthesized BH3-mimetics, ABT-737 and ABT-263, bound to both BCL2 and BCL-XL. Although ABT-263 (navitoclax) showed promising results in phase I studies, the dose limiting toxicity of the treatment was thrombocytopenia, as BCL-XL is essential for mature platelet survival [15]. ABT-199/VEN is a potent, selective, and orally available inhibitor of BCL2 [16], and its mechanism of action is shown in Figure 1. Preclinical studies have shown that VEN induced rapid apoptosis in a variety of AML cell line models at nanomolar concentrations and inhibited in vivo leukemia progression [17].

### 2.2. Main Clinical Studies Using Venetoclax in AML

VEN efficacy was assessed in a phase II single-agent study in R/R AML patients, showing a 19% overall response rate (ORR) and a median duration of remission of 48 days. Phase I/II studies combining VEN plus HMA or low-dose ARAC also showed a high response close to 65% and a favorable tolerance profile [18,19,20,21,22]. These results led to the registration randomized phase III trial “VIALE-A”, in which 431 patients were enrolled. Among them, 286 were allocated to the VEN-AZA group and 145 to the AZA group. The incidence of CR was higher with VEN-AZA than with the control regimen (36.7% vs. 17.9%; *p* < 0.001), as was the ORR = 66.4% vs. 28.3%; *p* < 0.001). After a median follow-up of 20.5 months, the median OS was 14.7 months in the VEN-AZA group and 9.6 months in the control group. Long-term follow-up confirmed the better OS in the VEN-AZA group [3,4]. The VEN-AZA regimen was approved for unfit or older AML patients in France and in most European countries in 2020. Since then, real-life experiences have confirmed the high response rate of VEN-AZA treatment in the context of newly diagnosed AML as well as refractory AML [23,24,25,26]. The main clinical studies published on VEN-HMA are summarized in Table 1.

## 3. Molecular Factors Modulating Venetoclax Efficacy

Current biological factors that are commonly used for AML in younger patients in the context of intensive chemotherapy (ICT) are questionable in the context of VEN-AZA treatment, as seen for instance in European Leukemia Network (ELN) 2022 cytogenetics and molecular classification [27]. Therefore, new molecular classifications are needed. In this section, we will review studies that have helped better patients’ stratification in the context of VEN-based regimens. The influence of the main AML gene mutations on VEN resistance are summarized in Figure 2.

### 3.1. IDH Mutations

Isocitrate dehydrogenase 1 and 2 (IDH1/2) are enzymes that perform key roles in various cellular functions, including the regulation of carbohydrate metabolism, epigenetics, and cell differentiation. Wild-type IDH1 and IDH2 oxidize and decarboxylate isocitrate to α-ketoglutarate (α-KG) in the cytoplasm and mitochondria, respectively, and simultaneously reduce NADP^+^ to NADPH. *IDH1/2* gene mutations (*IDH1/2*mut) are almost always heterozygous and occur in enzymatically active sites. Hotspot mutations in *IDH1*^R132^, *IDH2*^R140^, and *IDH2*^R172^, enable a neomorphic reaction that converts α kg into the oncometabolite *D*-2-hydroxyglutarate (*D*-2-HG), which competitively inhibits α-KG-dependent dioxygenases and induces an AML blast differentiation blockade [28]. 

The favorable impact of *IDH1/2* mutations in VEN-AZA-treated AML was already noticed in the phase II efficacy trial. Further analyses pooling results from VIALE-A and phase II studies showed 79% CR/CRi rates in *IDH1/2*mut patients. Interestingly, the response rates in the *IDH2*mut subgroup were higher (89%) than in the *IDH1*mut group (66.7%). *IDH1/2*mut were also associated with a longer median OS (24 months for *IDH1/2* patients vs. 14 months in general). Again, OS was better in the *IDH2*mut subgroup compared to the *IDH1*mut [29]. The median OS was improved in the group of patients with negative minimal residual disease (MRD) assessed by flow cytometry (42 months) [30]. Mechanistically, the effect of VEN on *IDH1/2*mut preclinical models has been related to the inhibition of mitochondrial cytochrome c oxidase by *D*-2-HG, causing a reduction in the mitochondrial threshold for apoptosis induction [31] as well as a decrease in mitochondrial respiration [11,32].

### 3.2. TP53 Mutation

In a recent actualization of the International Consensus Classification (ICC) 2022 AML and the World Health Organization (WHO) 2022 classification, *TP53* mutant (*TP53*mut) AML and myelodysplastic syndromes (MDS) have been individualized based on their poor prognosis [33,34]. Mutations in the *TP53* genes are presents in 10–15% of AML cases and are enriched in poor-risk AML because of frequent additional cytogenetics or molecular alterations [35]. Most *TP53*mut are missense mutations that involve the DNA-binding domain. Mutant premature termination codons and frameshift mutations as well as most single amino-acid substitutions or deletions result in a strong disruption of TP53 function.

*TP53*mut diseases are known to poorly respond to ICT, as reported in recent large cohort studies. The overall survival rates of 6–9 months are among the worst within AML subsets [36]. The results seen with VEN-AZA have also been deceiving. In a pooled analysis of phase II and III studies, the response rate was 70% in the absence of *TP53*mut vs. 41% in the presence of *TP53*mut, and median OS was 23.4 months in the absence of *TP53*mut vs. 5.2 months in the presence of *TP53*mut. Patients treated with VEN-AZA were not doing better than those treated with AZA monotherapy. These data were confirmed in a real-life study from the MD Anderson Cancer Center showing a 6–9-month OS [37,38] and also in a study with VEN and decitabine, in which patients with *TP53*mut had a 5.2-month OS [39]. Taken together, these data question the rationale of using VEN in this setting, in particular for fragile or elderly patients [40]. 

From a mechanistic perspective, loss of function mutations in the *TP53* gene decreased the expressions of pro-apoptotic genes such as NOXA or PUMA, which are known to be a transcriptional target of TP53 [41], as well as induced a less intense functional activation of the pro-apoptotic BAX and BAK proteins [42]. Consistently, studies using Crispr-Cas9 screens identified *BAK*, *BAX*, *PUMA*, and *NOXA*, as well as *TP53* target genes as crucial regulators of VEN activity in vitro [41,42,43,44,45,46]. A recent study showed that interactions between TP53 and increased MYC transcriptional program led to VEN resistance that could be targeted [47]. 

### 3.3. Signaling Mutations (Including FLT3 Mutations)

Fms-like tyrosine kinase 3 (*FLT3*) is mutated in approximately 30% of adult AML cases, including internal tandem duplication (*FLT3*-ITD) and tyrosine kinase domain (TKD) amino acid substitution (*FLT3-*TKD), leading to constitutive tyrosine kinase activity and constitutive activation of the downstream proliferative signaling cascade [48]. The negative impact of *FLT3*-ITD on AML patients’ survival when treated with ICT is well established, as are other signaling mutations such as *RAS* mutations [49]. 

The impact of signaling mutations is unclear in the context of VEN-AZA treatment [14,20,21,50,51]. The response rates and OS were similar in *FLT3*-ITD versus *FLT3*wt cohorts in a pooled analyses of 353 patients from the VIALE-A and phase II studies, including 42 with *FLT3* mutations [50]. Nevertheless, refined prognostic risk signatures based on *N/KRAS*, *FLT3*-ITD, and *TP53* mutations have shown that patients with signaling mutations have at least an intermediate risk [27,52,53]. Multiple small-molecule tyrosine kinase inhibitors that target FLT3 are in development in combination with VEN for the treatment of patients with elderly/unfit AML patients [48]. Some of these molecules represent interesting candidates to test in triplet therapies based on the VEN-AZA backbone. 

### 3.4. Secondary Type Mutations

The WHO 2022 and ICC 2022 classifications include a new AML subtype classifying AML with the presence of any of eight gene mutations, including *ASXL1*, *BCOR*, *EZH2*, *STAG2*, *SF3B1*, *SRSF2*, *ZRSR2*, and *U2AF1* (as well as *RUNX1* in the ICC 2022 classification only), as AML with myelodysplasia-related gene mutations (MR genes) [33,34,54]. MR gene mutations are associated with a poor response to ICT. As a result, the ELN 2022 classifies these AML cases in the poor-risk group [55]. Nevertheless, the impact of these gene mutations are less clearly defined in the context of VEN-AZA treatment [53,56]. For example, *ASXL1* mutations confer VEN sensitivity in preclinical models but not in clinical studies [57,58]. Mutations in splicing factors, including *SRSF2*, *U2AF1*, *SF3B1*, and *ZRSR2*, have been implicated in the pathogenesis of MDS and AML [59,60]. These mutations are encountered in approximately 50% of secondary AML cases and usually correlate with inferior outcomes to ICT [49,61,62]. In the context of VEN-treatment, the negative impact of these genes on outcomes is less clear, as some studies report the absence of a negative impact [63,64], while others report that SRSF2 mutation was associated with a poorer response in preclinical [65] and clinical studies [53,66].

### 3.5. Mutations in Apoptotic Genes

Mutations in the *BCL2* gene, in particular the BCL-2-binding groove variants G101V or D103Y, were shown to confer resistance to chronic lymphocytic leukemia (CLL) patients treated with VEN [67]. Contrary to CLL patients, *BCL-2* variants were not found in VEN R/R AML cohorts. Nonetheless, *BAX* deficiency was identified in in vitro studies based on genome-wide Crispr screens exploring VEN resistance [44]. BAX mutations were identified in normal hematopoietic stem and progenitor cells of CLL patients treated with VEN [68]. Consistently, mutations in the BAX gene were found in 7 of 41 patients (17%) at relapse after VEN-AZA treatment, with no cases identified among patients with primary refractory disease [69]. The *BAX* variants that emerged at relapse included frameshift, nonsense, or donor splice-site abnormalities upstream of the BAX COOH-terminal domain or just before the BAX COOH-terminal domain, disrupting the COOH-terminal α9 helix necessary for BAX translocation from the cytosol into the mitochondrial outer membrane. Further studies will help us to understand the differential mechanisms of resistance between CLL and AML patients treated with VEN.

## 4. Non-Genetic Factors Driving Venetoclax Efficacy

### 4.1. Regulators of Intrinsic Apoptosis

Intrinsic apoptosis induction relies on the balance between the anti-apoptotic proteins (BCL-2, BCL-XL, MCL-1, BCLW, and BFL-1) and the pro-apoptotic proteins, which are divided into two groups: the effectors (BAX, BAK, and BOK) and the BH3-only proteins (BIM, BAD, NOXA, PUMA, BID, BIK, HRK) [70]. Consequently, both the upregulation of anti-apoptotic and the downregulation of pro-apoptotic signals can modulate AML cells’ sensitivity to BH3-mimetics and induce VEN resistance.

(a)Increase in anti-apoptotic proteins.

BCL2

Several studies have shown that BCL2 is often overexpressed in AML cells, and is associated with poor prognosis and resistance to chemotherapy [71,72,73]. This phenomenon has also been found with VEN in vitro. Mechanistically, overexpression of BCL2 is related to its phosphorylation both in myeloid and lymphoid malignancies [74,75,76]. As a matter of fact, BCL2 dependency seems to be more important than expression level. Such a dependency on anti-apoptotic proteins can be revealed by performing a *“*BH3 profiling”, assessing cancer cell susceptibility to induce MOMP. In this assay, AML cells are treated in vitro with synthetic BH3 peptides that selectively interact with pro-survival BCL-2 family proteins. For example, dependencies on BCL2 or MCL1 are inferred if MOMP occurs in response to BAD or MS-1 peptides, respectively [77,78]. An elegant study from the Letai’s lab showed that BH3 mimetic resistance is characterized by decreased mitochondrial apoptotic priming, both in PDX models and human clinical samples, due to alterations in BCL2 family proteins that vary among cases independently of acquired mutations in leukemia genes [79].

MCL1

Upregulation of the other anti-apoptotic MCL1 have also been identified as major determinants of resistance to intensive chemotherapy [80] as well as VEN resistance in both lymphoid and myeloid leukemias [76,81]. A strong correlation between MCL1 expression/dependency and abnormalities in signalization pathways, such as *FLT3* mutations, AKT, or RAS/MAP deregulation, has been observed [82,83,84,85,86]. Moreover, differentiation status influences both signalization and MCL1 dependency. For instance, differentiation state and BH3 family change in protein expression are highly dependent [87,88,89,90,91]. Several studies have shown that monocytic AML is more resistant to VEN, and treatment with VEN selected differentiated monocytic cells at relapse [88,90].

BCLXL and BCL2A1

A recent study spotlighted on the functional dependence of AML subtypes’ erythroid or megakaryocytic differentiation, including acute erythroid leukemia (AEL), acute megakaryoblastic leukemia (AMKL), and pro-survival B-cell lymphoma-extra-large (BCL-xL) dependencies. Interestingly, BCLXL expression in relation to erythroid differentiation is highly connected with *TP53* status, which is frequently mutated in these diseases [92,93]. In other studies, transcriptomic analysis of AML patients’ samples showed differential expressions of BCL2A1 in VEN-resistant cells. Upregulation of BCL2A1 expression has been correlated with higher VEN resistance. Knockdown of BCL2A1 restored apoptosis and reduced cell growth in the resistant AML cells without any substantial effect on the CD34+ hematopoietic stem and progenitor cells [65,94].

(b)Decrease in pro-apoptotic signals.

Crispr screens studies have identified NOXA and PUMA as critical regulators of VEN activity in vitro [42,44,45,46]. Consistently, resistant AML cells have often been associated with a decrease in the expression of pro-apoptotic BAX and NOXA [75,95]. Mechanistically, it is noticeable that TP53 transcriptionally regulates the expression of BAX and NOXA proteins [41,43,96,97,98], partially explaining the role of *TP53* mutation in mediating VEN resistance. As a proof of concept, NOXA-deficient cells were shown to be resistant to the combination of VEN-AZA and PUMA in vitro [99], which epigenetically mediated deregulation, causing VEN resistance in lymphoid [100] and myeloid leukemias [101].

### 4.2. Differentiation Status

Classical models have shown normal and pathologic hematopoiesis as a hematopoietic differentiation cascade with distinct cell states between stem/progenitor and terminally differentiated cells. It is now admitted that leukemic differentiation is more complex, based on single-cell analyses showing clonal heterogeneity [10,102]. An analysis of the BEAT AML data set (NCT03013998) demonstrated increased VEN resistance among leukemic patients with M4 and M5 AML [103]. Single-cell multiomics gave a better understanding of how monocytic differentiated subclones impact resistance to BCL2 inhibition in AML [90]. Differentiation status is somehow related to various expressions of pro- and anti-apoptotic protein. A recent study found that AML leukemic stem cells (LSC), based on flow cytometry sorting, were eliminated by VEN-AZA and determined the therapy outcome. Authors have developed and validated a score linking the ratio of protein expression of anti-apoptotic proteins BCL2, BCLXL, and MCL1 in LSCs to VEN response [104]. Together with ex vivo sensitivity profiling testing [91,92,105], these new tools would help to better identify resistance at a single-cell level.

### 4.3. Metabolism and Beyond

Mitochondria are organelles closely linked to cell death and cellular respiration, as well as other metabolic pathways. Crispr screen studies have found many genes that related to mitochondrial metabolism or structure critically regulate drugs targeting mitochondria and VEN sensitivity in vitro [45,46,106]. A recent example is *CLPB*, a gene maintaining the mitochondrial cristae structure via its interaction with the cristae-shaping protein OPA1. The CLPB protein is upregulated in human AML and is further induced upon acquisition of VEN resistance. Consequently, its ablation sensitizes AML to VEN [45].

Work from our lab has found that targeting mitochondrial iron can induce a dramatic loss of mitochondrial respiration, inducing BAX/BAK activation through pathways independent of BH3 mimetics. This new cell death modality on the frontier between apoptosis and ferroptosis was highly synergistic with VEN in vitro and in vivo [106]. Other studies have underpinned the important role of mitochondrial chaperones such as MARCH5 E3 ubiquitin ligase [107,108] or DRP1 [109].

Finally, several studies have tried to better understand the role of OXPHOS [110], mitochondrial respiration [111], amino acids [112], and lipids such as fatty acids [113,114,115,116], nicotinamide [117], and ceramide [118,119], in VEN response. It is likely that a better understanding of the integrated stress response (ISR) mediated by the ATF4 transcription factor will help to fill the gap between cell death, metabolism, and VEN resistance [46,91,106,120].

Another layer of complexity involves the role of the microenvironment. It is noteworthy that VEN enhances T cell-mediated antileukemic activity by increasing ROS production [121] and that MCL1 targeting can modulate leukemia cell metabolism, cell adhesion proteins, and leukemia–stromal interactions [122].

## 5. Overcoming Resistance with New Therapies

### 5.1. Hypomethylating Agents

The rationale for the synergistic activity between BCL2 inhibitors and AZA has been explored in preclinical works. It has been shown that AZA does not affect the expression of anti-apoptotic proteins but induces the expression of the pro-apoptotic BH3 proteins NOXA and PUMA only, resulting in a priming of AML cells to intrinsic apoptosis. Interestingly, this phenomenon is not epigenetically regulated, as shown by the absence of methylation changes in DNA promoters and the rapid kinetics of apoptosis induction [99,123]. This synergy may also be related to the metabolic activity of VEN-AZA on the leukemia stem cell [7,112]. The optimization of current non-intensive regimens such as weekly decitabine [124] and the use of alternative oral hypomethylating drugs may possibly modulate the synergistic effects obtained with VEN [125,126,127]. 

### 5.2. Innovative BH3 Mimetics

MCL1

There is a strong rationale for targeting MCL1 anti-apoptotic protein in preclinical models based on dependency of AML cells for sustaining blast proliferation [42,75,79,128,129]. Nevertheless, clinical development was halted by cardiac toxicity (mainly troponin elevations). Current MCL1 trials assess the safety of these MCL1 inhibitors as single agents, with the provision of combination therapy with VEN (NCT03672695, NCT02675452, NCT03218683) [130]. Recent reviews have covered the field of MCL1 inhibitors currently in development [131,132,133].

Dual BCL2 and BCLXL inhibitors

Despite its toxicity to platelets, navitoclax has seen renewed interest in targeting resistant AML blasts. The phase II TRANSFORM II study (NCT04472598) is currently recruiting relapsed or refractory myelofibrosis patients. New dual BCL2/BCLXL inhibitors are in development, such as AZD4320 [134]. VHL-recruiting proteolysis-targeting chimeras (PROTACs) cause BC2 and BCLXL degradation via an interaction with the von Hippel–Lindau E3 ubiquitin ligase. As platelets lack VHL expression, the drug spares the on-target platelet toxicity caused by navitoclax [135].

### 5.3. Tyrosine Kinase Inhibitor (TKI)

Indirect MCL-1 inhibition can be achieved by targeting cyclin-dependent kinases, including 7 and 9. Some early trials show promising responses [136]. The CDK9 inhibitor AZD4573 was tested in R/R AML (NCT05140382) and showed a 15% response rate in this heavily pretreated AML patients population [137]. Other cyclins may be of interest, such as CDK7/12/13 [138]

### 5.4. FLT3 Inhibitors

There is extensive preclinical evidence in combining VEN and FLT3 inhibitors (FLT3inh), including gilteritinib, midostaurin, and quizartinib [139,140,141]. Gilteritinib is a type 1 FLT3 inhibitor which has activity against FLT3, ALK, and AXL [48]. Mechanistically, VEN with gilteritinib decreased phosphorylation of ERK and GSK3B via combined AXL and FLT3 inhibition, with subsequent suppression of the anti-apoptotic protein MCL1 [142]. The combination of VEN plus gilteritinib in *FLT3*mut patients was shown to induce a CR rate of 75% and median OS of 10 months in relapsed or refractory AML patients who had received prior FLT3 inhibitor therapy in 64% [143]. A study exploring giteritinib combined with VEN-AZA is also recruiting patients in a phase I/II trial [144].

Midostaurin acts both on *FLT3*-ITD and *FLT3*-TKD, and has a broad activity against tyrosine kinase receptors including PKC-α, FLT3, c-KIT, VEGFR, and PDGFR [48]. Midostaurin combined with VEN-AZA seemed to be feasible based on subset analyses from multiple investigator-initiated clinical studies, with a high response rate but significant hematological toxicities [145,146].

Quizartinib shows high selectivity to FLT3, KIT, colony-stimulating factor-1 receptor, platelet-derived growth factor receptor, and RET kinase [48]. In a phase I/II study presented at the ASH meeting in 2023, 50 patients were treated with a triplet combining VEN, decitabine, and quizartinib. This series included 40 R/R and 10 newly *FLT3*-ITD-mutated AML patients. Among the R/R patients, the composite CR rate (including CR and CR with incomplete hematological recovery) was 68%. The 60-day mortality rate was 0%. With a median follow-up of 33 months, the median OS was 7.1 months. Non-hematologic toxicities included pneumonia (73%), neutropenic fever (53%), sepsis (18%), bacteremia (15%), and other infections involving the skin (18%) and the gastrointestinal tract (13%) [147].

### 5.5. IDH Inhibitors

The efficacy of combining AZA with the IDH1 inhibitor ivosidenib was shown in the phase III AGILE study. This regimen is now approved for newly diagnosed *IDH1*mut AML patients [148]. Given the high response rate of *IDH*mut patients, the triplet regimen was further evaluated. A phase I/II trial showed a promising 90% composite complete response rate and 42 months median overall survival [149]. The safety and efficacy of combining the IDH2 inhibitor enasidenib with VEN-AZA was also evaluated in AML patients with *IDH2*mut. The response rates were also very high [150]. The benefit from adding IDH inhibitors to VEN-AZA must be confirmed in randomized trials. The main classes of drugs found to synergize with VEN in clinical studies are summarized in Figure 3.

### 5.6. Future Therapies

Targeting the interaction between tumor suppressor P53 and the E3 ligase MDM2 represents an attractive treatment approach for cancers with wild-type or functional TP53. Indeed, several small molecules have been developed and evaluated in various malignancies [151]. The MDM2 inhibitors developed in clinics include idasanutlin and milademetan (DS3032b). These drugs are currently in phase I trials in combination with VEN for *TP53*wt patients [152,153].

For *TP53*mut patients, few therapeutic advances have been made so far. Eprenetapopt (APR-246) is a first-in-class, small-molecule p53 reactivator. A phase I study combining the drug with VEN-AZA has associated it with an acceptable safety profile and encouraging activity. Indeed, 25 (64%) patients had an overall response and 15 (38%, 23–55) had a complete response [154].

Menin inhibitors constitute a novel class of agents targeting the underlying biology of nucleophosmin (NPM1) mutant and KMT2A rearranged (KMT2A acute leukemias [155,156]. Several menin inhibitors have been developed for AML, including revumenib, ziftomenib, and JNJ-6617. This last drug is currently being evaluated in monotherapy and in combination with VEN-AZA in two phase I clinical trials, 75276617ALE1001 (NCT04811560) and 1002, respectively [157].

Innovative immunotherapies are under evaluation, including the monoclonal antibody magrolimab against CD47. Development of the drug is currently on hold because of its lack of efficacy [158]. Nevertheless, the Enhance-3 study including newly diagnosed AML cases displayed promising results (80% overall response rate) in the frontline setting [159]. The antibody–drug conjugate targeting CD123 IMGN632, also known as pivekimab sunirine (or PVEK), is currently being tested in monotherapy or with VEN-AZA for patients with CD123-positive AML (NCT04086264). The first results were presented both in R/R and frontline cohorts and showed a tolerable safety profile and promising efficacy [160].

The main clinical studies of triplet therapies, including VEN-AZA plus other drugs, are summarized in Table 2.

## 6. Conclusions

Non-intensive approaches are rarely curative in AML. Historically, HMA monotherapy had been associated with a 20–30% response rate and a 10-month median OS. The development of VEN-HMA-based combination regimens has revolutionized the treatment of these aggressive diseases, as it has led to a doubling of response rates and a significant increase in survival. Despite these good results, relapses and primary refractory AML are still a matter of concern.

It is now clear that TP53 mutations are associated with the lack of efficacy of VEN-AZA regimens contrary to *IDH1/2* mutations (in particular *IDH2*), which confers the high efficacy of VEN. For most patients who are not harboring these molecular alterations or for those who harbor multiple alterations, the efficacy of VEN-AZA regimen is hard to anticipate. The utilization of minimal residual disease (MRD) assessments by flow cytometry and molecular analyses, although well described in the context of ICT [161,162], have to be clearly defined to guide therapeutics and prevent relapse upon VEN-AZA use [163,164,165]. It is likely that single-cell DNA sequencing using, for instance, the Tapestri^®^ plateform developed by mission bio [166] will help us to better understand the role of clonal evolutions in the context of resistance.

Triplet therapies based on the combination of VEN/HMA + targeted agents may represent good options for patients with targeted molecular alterations, including *IDH*, *FLT3,* and *NPM1* or *KMT2A* alterations; however, to date, evidence is lacking for supporting the use of triplets instead of VEN-AZA bi-therapy. It is likely that a combination with IDH or menin inhibitors will be feasible in the context of elderly AML patients given the low apparently hematological toxicities that have been reported with these two classes. On the other hand, combinations with FLT3 inhibitors may be difficult given the high additional toxicity of these classes of drugs with VEN-AZA. Therefore, these types of regimens may be reserved for fit patients failing intensive chemotherapy.

Finally, in the context of elderly AML, quality of life preservation is a major goal to achieve. Some retrospective studies have shown the feasibility of stopping VEN-AZA in remission [167,168]. It is likely that in the future, VEN-AZA regimens will be a backbone for new triplet therapies and off-treatment periods based on precise molecular alteration assessments and single-cell MRD follow-up.

## Figures and Tables

**Figure 1 cancers-16-01091-f001:**
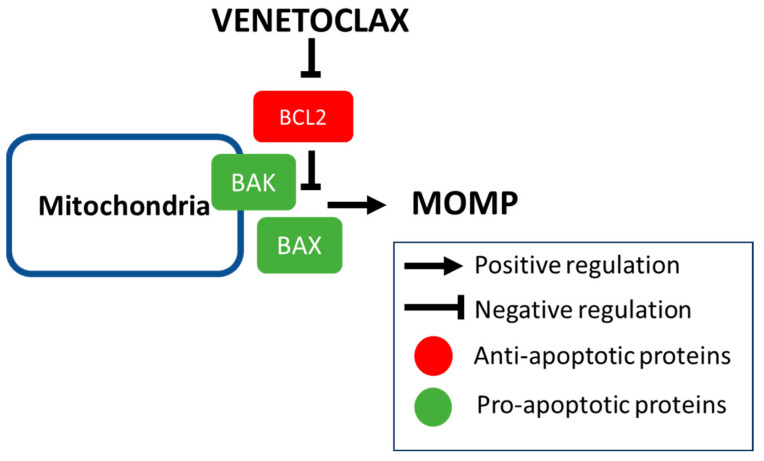
Mechanism of action of venetoclax. Venetoclax acts as a protein/protein interaction inhibitor that specifically binds the anti-apoptotic protein BCL2 and releases the pro-apoptotic proteins including BAX and BAK that induce mitochondrial outer membrane permeabilization (MOMP) and cell death by intrinsic apoptosis.

**Figure 2 cancers-16-01091-f002:**
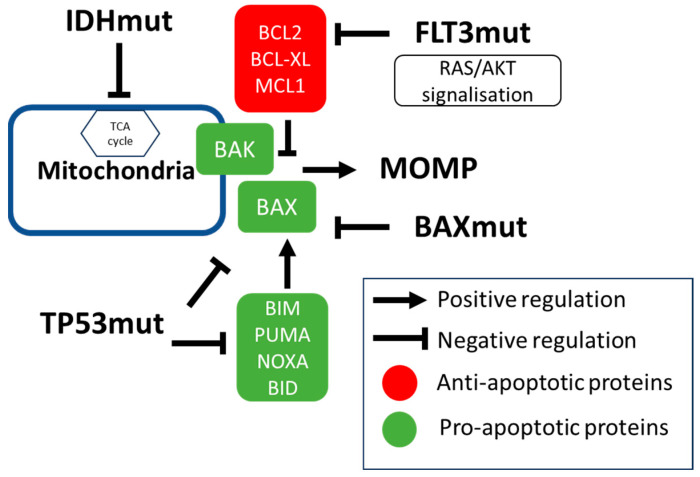
Main genetic alterations influencing VEN resistance. Intrinsic apoptosis relies on the balance between pro- and anti-apoptotic proteins that will induce mitochondrial outer membrane permeabilization (MOMP). In response to apoptotic stimuli, BAX and BAK form pores in the mitochondrial membrane, inducing mitochondrial outer membrane permeabilization (MOMP). This phenomenon is increased by the members of the BCL2 protein family containing a single BH3 domain named “BH3-only proteins”, which have a pro-apoptotic role (BIM, NOXA, PUMA, BID). Conversely, MOMP is blocked by a series of proteins that have an anti-apoptotic role including BCL2, BCL-XL, and MCL1. *IDH* mutations are associated with a higher sensitivity to VEN mostly by their role in mitochondrial metabolism and the tricarboxylic acid cycle (TCA). *TP53* mutations confer resistance to VEN by transcriptionally decreasing pro-apoptotic proteins such as NOXA or PUMA. *FLT3* mutations are associated with VEN resistance because of increased signalization unbalancing BCL2 and MCL1 dependencies. *BAX* mutations disrupt the COOH-terminal α9 helix necessary to BAX translocation from the cytosol into the mitochondrial outer membrane, decreasing MOMP.

**Figure 3 cancers-16-01091-f003:**
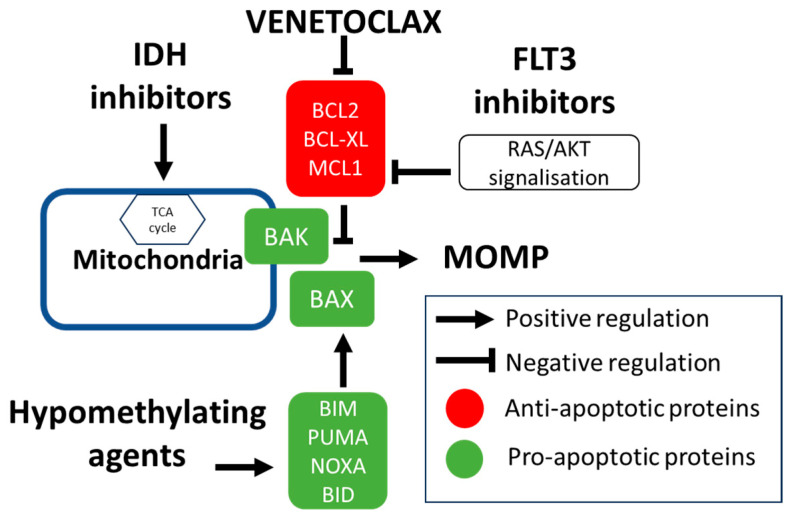
Main classes of drug synergizing with venetoclax in clinical studies. Hypomethylating agents such as azacitidine have been shown to induce an integrated stress response, upregulating the transcription of pro-apoptotic proteins NOXA and PUMA, which increases the efficacy of BCL2 inhibition. IDH inhibitors act by blocking the formation of the oncometabolite *D*-2-HG in IDH-mutated AML. IDH-mutated AML has been shown to be more susceptible to intrinsic apoptosis due to defects in the TCA cycle. FLT3 mutations induce MCL1 protein upregulation through the RAS/AKT signalization cascade. FLT3 inhibitors are deemed to cause an indirect MCL1 inhibition by degrading MCL1.

**Table 1 cancers-16-01091-t001:** Main studies evaluating VEN-HMA in the context of newly diagnosed AML patients ineligible for intensive chemotherapy.

Phase; Name; ID	Study Population	Venetoclax Doses	Other Agent Administration Regimens	Response Rate	Survival Rate
I; (M14-358); NCT02203773	ND AML ineligible for chemotherapy	400–800–1200 mg D1 to D28	AZA 75 mg/m^2^, D1 to D7 or DEC 20 mg/m^2^, D1 to D5	37% (CR), 68% (ORR), 83% (cCR)	17.5 months (median OS)
II; NCT03404193	ND patients with AML > 60 years	400 mg D1 to D28 (D1 to D21 if blasts < 5%)	DEC 20 mg/m^2^, D1 to D10 (induction) DEC 20 mg/m^2^, D1 to D5 (consolidation)	63% (CR + CRi), 74% (ORR)	18.1 months (median OS)
III; (VIALE-A); NCT02993523	ND AML ineligible for chemotherapy	400 mg D1 to D28 vs. placebo	AZA 75 mg/m^2^, D1 to D7	36.7% vs. 17.9% (CR), 64.7% vs. 22.8% (CR + CRi)	14.7 months vs. 9.6 months (median OS)

Abbreviations: AZA, azacitidine, cCR, composite complete response; CR, complete response; CRi, CR with incomplete hematological recovery, DEC, decitabine; MLFS, morphologic leukemia free state; ND, newly diagnosed; ORR, overall response rate; OS, overall survival.

**Table 2 cancers-16-01091-t002:** Main studies evaluating VEN-AZA-based triplet therapies.

Treatment	Study ID	Phase	Number of Patients	Response	Treatment
Quizartinib (FLT3 inhibitor)	NCT03661307	I/II	28 (23 R/R AML FLT3mut)	78% achieved cCR (3 CR, 15 CRi)	[147]
Ivosidenib (IDH1 inhibitor)	NCT03471260	Ib	31 (all R/R, IDHmut)	94% had a CR + CRh + CRi + PR + MLFS	[149]
Magrolimab (anti CD47)	NCT04435691	I/II	60 (41 ND, 29)	ORR [CR, CRi, MLFS] Frontline = 80%	[159]
Pivekimab sunirine (PVEK, anti CD123)	NCT04086264	I/II	50 ND patients	CR rate was 52% (26/50), and cCR rate was 66% (33/50)	[160]

Abbreviations: cCR, composite complete response; CR, complete response; CRi, CR with incomplete hematological recovery, MLFS, morphologic leukemia free state; ND, newly diagnosed; ORR, overall response rate, PR, partial response.

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
