# Peer review of "Venetoclax Resistance in Acute Myeloid Leukemia"

_cancers, 2024, doi:10.3390/cancers16061091_

Round 1

Reviewer 1 Report

Comments and Suggestions for Authors

In this review, the Authors have exhaustively described mechanisms of action and resistance to anti-BCL-2 inhibitors in acute myeloid leukemia (AML).

Including a figure for mechanisms of action and for section 5 would be helpfull to improve the manuscript.

Some data on real-life experience should be reported in 2.3 section.

Comments on the Quality of English Language

Few typos present

Author Response

Including a figure for mechanisms of action and for section 5 would be helpfull to improve the manuscript.

We thank the reviewer for this suggestion. We added a new Figure 1 describing the mechanism of action of venetoclax and a new figure 3 summarizing the drugs found to be synergistic with venetoclax in clinical studies. We hope these new figures will improve the manuscript.

Some data on real-life experience should be reported in 2.3 section.

We thank the reviewer for this important comment. We added a sentence as well as 4 additional references on lines 95-97. “Since then, real life experiences have confirmed the high response rate of VEN-AZA treatment in the context of newly-diagnosed AML as well as refractory AML [23], [24], [25], [26]”.

Reviewer 2 Report

Comments and Suggestions for Authors

The treatment of acute myeloid leukemia in old patients still remains changeling. It has been largely proven that intensive chemotherapy did not improve response rate and survival. So, in the past few years, the combination of Hypomethylating agents and  the BCL2 inhibitor venetoclax, has  almost completely replaced the standard “3+7” chemotherapy, improving response rate with acceptable toxicity. 

However, primary resistance to venetoclax and relapses are frequent, and many genetic alterations, as well as non-genetic factors has been hypothesized to explain response failure.

In this paper the authors provide a concise, but precise review of factors able to impair HMA/VEN efficacy and summarized the current tools potentially able to overcome resistance.

The paper is clear, as well the figure and references are adequate. 

I have just a question:

·      How can we predict efficacy /resistance of HMA/VEN in patients candidate for non-intensive chemotherapy? How the concomitant presence of  more than one factor affects HMA/VEN efficacy? 

Author Response

I have just a question:

  • How can we predict efficacy /resistance of HMA/VEN in patients candidate for non-intensive chemotherapy? How the concomitant presence of  more than one factor affects HMA/VEN efficacy? 

We thank the reviewer for these important questions. We added a sentence in the last section of the manuscript discussing the possible impact of minimal residual disease using DNA sequencing at a single cell level in this context (lines 467-474). We hope this explanation will improve the quality of the manuscript. “For most patients, who are not harboring these molecular alterations or who harbor multiple alterations, efficacy of VEN-AZA regimen is hard to anticipate. The utilization of minimal residual disease (MRD) assessment by flow cytometry and molecular analyses although well described in the context of ICT [162], [163] has to be clearly defined to guide therapeutics and prevent relapse upon VEN-AZA [164], [165], [166]. It is possible that single cell DNA sequencing using for instance Tapestri ® plateform developed by mission bio [167] will help to better understand the role of clonal evolutions in the context of resistance.”

Reviewer 3 Report

Comments and Suggestions for Authors

Venetoclax has shown promise as a potential therapy for AML patients who are unable to receive induction chemotherapy, regardless of whether they are newly diagnosed, relapsed, or refractory. However, response rates are highly dependent on patient characteristics and mutational profiles. Ongoing clinical trials are exploring the possibility of using it as a frontline treatment for all AML patients.

In their review, Garciaz et al. explored the issue of Venetoclax resistance in AML treatments, examining the molecular factors that may contribute. However, the article lacked originality and did not offer any new insights, merely rehashing information that had already been discussed in previous reviews. As a result, it may be less engaging for readers.
To make the review more impactful, the author could have provided clear take-home messages for readers, helping them to better understand the current state of AML treatment and the direction it is headed in. By offering concrete suggestions, the article would have been more persuasive, leaving readers with a sense of inspiration and motivation.
Here are some suggestions to help the author improve their review:
Simple Summary: Please double-check the spelling and grammar of the text.

Introduction: Consider using a schematic representation to help readers quickly grasp the discussion.
Main Body: Include a table that provides an overview of studies using Venetoclax-based therapy for AML patients. Additionally, it would be helpful to include common adverse events associated with AML therapies. 
Discuss sensitivity and resistance to Venetoclax by breaking down patients into molecular-defined subgroups.
Take-Home Message: In this section, discuss your predictions for active Venetoclax-based combination therapies for AML in the future. Your thoughts and ideas here will have a lasting impact on readers.

Comments on the Quality of English Language

Some spelling mistakes need to be rechecked.

Author Response

Here are some suggestions to help the author improve their review:
Simple Summary: Please double-check the spelling and grammar of the text.

We would like to thank the reviewer for this comment. We have double checked the Simple summary.

Introduction: Consider using a schematic representation to help readers quickly grasp the discussion.

We thank the reviewer for this comment. We have added a new Figure 1, describing the mechanism of action of venetoclax.

Main Body: Include a table that provides an overview of studies using Venetoclax-based therapy for AML patients. Additionally, it would be helpful to include common adverse events associated with AML therapies.

We thank the reviewer for this comment. We have added a new Table 1, describing the most important clinical studies using VEN-HMA for newly diagnosed AML.

Discuss sensitivity and resistance to Venetoclax by breaking down patients into molecular-defined subgroups.

We thank the reviewer for this comment. We split patients into molecular subgroups including TP53, IDH, signaling mutations, secondary type mutations and mutations in the apoptotic machinery in the section 3 of the manuscript.

Take-Home Message: In this section, discuss your predictions for active Venetoclax-based combination therapies for AML in the future. Your thoughts and ideas here will have a lasting impact on readers.

We thank the reviewer for this suggestion that may improve the interest of the readers. We write a new “take home paragraph including our thoughts regarding the use of MRD, the management of triplets therapies and the evenutallity of treatment cessation for responding patients in the future from line 466 to line 489.

“It is now clear that TP53 mutations are associated with a lack of efficacy of VEN-AZA regimen contrary to IDH mutations (in particular IDH2) which confers a high efficacy of VEN. For most patients, who are not harboring these molecular alterations or who harbor multiple alterations, efficacy of VEN-AZA regimen is hard to anticipate. The utilization of minimal residual disease (MRD) assessment by flow cytometry and molecular analyses although well described in the context of ICT [162], [163] has to be clearly defined to guide therapeutics and prevent relapse upon VEN-AZA [164], [165], [166]. It is possible that single cell DNA sequencing using for instance Tapestri ® plateform developed by mission bio [167] will help to better understand the role of clonal evolutions in the context of resistance.

Triplet therapies based on the combination of VEN + HMA + targeted agents may represent good options for patients with targeted molecular alterations, including IDH, FLT3 and NPM1 or KMT2A alterations but to date evidence is lacking for supporting the use of triplets instead of VEN-AZA bi-therapy. It is likely that combination with IDH or Menin inhibitors will be feasible in the context of elderly AML patients given the low apparent hematological toxicities that has been reported with these two classes. On the other hand, combination with FLT3 inhibitors may be difficult given the high additional toxicity of these class of drugs with VEN-AZA. Therefore, these types of regimens may be reserved for fit patients failing intensive chemotherapy.

Finally, in the context of elderly AML, quality of life preservation is a major goal to achieve. Some retrospective studies have shown the feasibility of stopping VEN-AZA in remission [168], [169]. It is likely that in the future, VEN-AZA regimen will be a backbone for new triplet therapies and off-treatment periods based on precise molecular alterations assessment and single cell MRD follow up.”

Round 2

Reviewer 3 Report

Comments and Suggestions for Authors

Thanks to all the authors for there great work on my review.